# A 3D Plasmonic Crossed-Wire Nanostructure for Surface-Enhanced Raman Scattering and Plasmon-Enhanced Fluorescence Detection

**DOI:** 10.3390/molecules26020281

**Published:** 2021-01-08

**Authors:** Chun-Ta Huang, Fuh-Jyh Jan, Cheng-Chung Chang

**Affiliations:** 1Protrustech Co., Ltd., 3F.-1, No.293, Sec. 3, Dongmen Rd. East District, Tainan City 701, Taiwan; win@protrustech.com; 2Department of Plant Pathology, National Chung-Hsing University, Taichung 402, Taiwan; 3Advanced Plant Biotechnology Center, National Chung Hsing University, Taichung 402, Taiwan; 4Graduate Institute of Biomedical Engineering, National Chung Hsing University, Taichung 402, Taiwan; 5Intelligent Minimally-Invasive Device Center, National Chung Hsing University, Taichung 402, Taiwan

**Keywords:** nanowire, crossed-wire woodpile, antenna effect, hot spot effects, plasmon-enhanced fluorescence (PEF), surface-enhanced Raman scattering (SERS)

## Abstract

In this manuscript, silver nanowire 3D random crossed-wire woodpile (3D-RCW) nanostructures were designed and prepared. The 3D-RCW provides rich “antenna” and “hot spot” effects that are responsive for surface-enhanced Raman scattering (SERS) effects and plasmon-enhanced fluorescence (PEF). The optimal construction mode for the 3D-RCW, based on the ratio of silver nanowire and control compound R6G, was explored and established for use in PEF and SERS analyses. We found that the RCW nanochip capable of emission and Raman-enhanced detections uses micro levels of analysis volumes. Consequently, and SERS and PEF of pesticides (thiram, carbaryl, paraquat, fipronil) were successfully measured and characterized, and their detection limits were within 5 μM~0.05 µM in 20 µL. We found that the designed 3D plasmon-enhanced platform cannot only collect the SERS of pesticides, but also enhance the fluorescence of a weak emitter (pesticides) by more than 1000-fold via excitation of the surface plasmon resonance, which can be used to extend the range of a fluorescence biosensor. More importantly, solid-state measurement using a 3D-RCW nanoplatform shows promising potential based on its dual applications in creating large SERS and PEF enhancements.

## 1. Introduction

Surface plasmon resonance (SPR) is a phenomenon based on collective oscillations of surface electrons in metallic nanostructures. The SPR character strongly depends on the noble metal species, size, and shape of the nanostructures [1]. Plasmon-enhanced optical sensors built using metallic nanostructures can be designed to detect analytes in various fields [2]. With respect to SPR, localized SPR (LSPR) can reveal nonpropagating oscillations of surface electrons, which can concentrate the incident electromagnetic (EM) field around the nanostructure. Thus, the local EM field can promote optical properties such as fluorescence to cause plasmon-enhanced fluorescence (PEF); and Raman scattering resulting in surface-enhanced Raman scattering (SERS), which can be several orders of magnitude stronger than the incident field strength [3]. On the other hand, LSPR occurs when the dimensions of a metallic nanostructure are less than the incident light wavelength. The plasmon energy (peak position) of LSPR will change with shape. A more redshifted plasmon will be observed in larger nanostructures [4]. The localized surface plasmon resonances (LSPR) accompanied by electromagnetic field enhancements exhibited by metallic nanostructures have found utility in photocatalysis [5,6], medical diagnostics [7,8], biological and molecular sensing [9,10], and surface-enhanced Raman scattering (SERS) [9,11,12].

From the perspective of morphology, it is known that in the low scale of a nanostructure, the energy of the LSPR is quickly converted into heat, which then leads to a strong absorption by electron-electron scattering. Alternatively, the electron-electron surface scattering is reduced in a larger nanostructure, and the energy of the plasmons is reradiated, leading to a strong scattering cross section [13], which means that the LSPR energy is reradiated into the far field as scattered radiation [14]. Furthermore, the nonspherical nanostructures can support anisotropic plasmons to drive larger SERS enhancements of analytes at their cross section (cross stacking), sharp tips, or edges. These hot-spot regions can possess an electric field amplitude that can be orders of magnitude larger, which leads to intense near-field EM and a much shorter decay than that found for typical SERS [15].

Followed by the discovery of SERS, plasmon-enhanced fluorescence (PEF) was soon characterized as one of the surface-enhanced spectroscopy techniques [16,17]. When excited, metal nanoparticles such as gold or silver nanoparticles commonly show a broad plasmon spectrum and the optical extinction cross section (absorption + scattering) can be several orders of magnitude higher than that of fluorophores [18]. The main factor in PEF is an increase in the sample’s absorption and emission cross sections, which is ascribed to the local field enhancement associated with the excitation of an LSPR in the metal nanostructure. On the other hand, once the fluorophore’s emission energy couples with a metal plasmon, it can cause the metal to radiate with enhanced intensity in situ as fluorophore luminescence, which is also called metal-enhanced fluorescence (MEF) [19]. The overlap between the LSPR of a metal nanoparticle and the molecular absorption and emission spectra for the fluorophore is predicted to yield the highest fluorescence enhancement factor [20]. PEF not only offers enhanced emission and a decreased lifetime but also allows an expansion of the field of fluorescence by incorporating weak quantum emitters, avoiding photobleaching, and providing the opportunity of imaging with resolutions significantly better than the diffraction limit. It also opens up a window to a new class of photostable probes by combining metal nanostructures and quantum emitters [21].

Silver metal (Ag) nanostructure materials such as nanoparticles (NPs) and nanowires (NWs) have attracted widespread interest due to their unique and tunable optical properties that arise mainly from an LSPR effect [22,23]. Especially, Ag NWs have a high aspect ratio for their length to their diameter and can, thus, be used to build blocks for fabricating two-dimensional and three-dimensional nanostructures. Such nanostructures can be constructed to form transparent, flexible, conductive 2D layers for use in flexible electronic devices as potential replacements for transparent conducting oxide films [24]. Ag NWs can also be assembled into 3D stacked plasmonic substrates for use in various sensing applications, including surface-enhanced Raman spectroscopy (SERS) and plasmon-enhanced fluorescence (PEF) sensors [25,26]. In a previous study, we proved that for either a small organic molecule or organic nanoparticle, a nanowire offers more apparent metal-enhanced fluorescence over a nanoparticle, and we built a double emission enhancement (DEE) sensor platform based on a nanowire-based chip [27].

This study provides a key design consideration for the use of hot spots for anisotropic nanostructures for SERS and PEF, which begins with the synthesis of Ag NWs and the fabrication of 3D Ag NWs on a nanostructure substrate to form a 3D random crossed-wire woodpile (3D-RCW). After identifying the conformation, we evaluated plasmonic properties of 3D-RCW according to the criteria of plasma generation and found that 3D multilayer stacks of Ag NWs can provide both in-plane LSPR coupling among the parallel NWs and out-of-coupling at the cross-points at which two nanowires are closely stacked. That is, 3D stacked Ag NWs enable concentrated plasmon at closed-packed Ag NW structures, which is several orders of magnitude higher than that of the 2D substrate. Therefore, the local electromagnetic (EM) field can amplify the Raman scattering of adsorbed molecules and mediate fluorescence in fluorescent species. Here, we successfully applied 3D-RCW to the plasmon-enhanced spectroscopic techniques SERS and PEF to execute a variety of chemical and biological sensing applications. The related principles and protocols of 3D-RCW used for SERS and PEF conditions are listed in the Section 2 and in Figure 1.

## 2. Experiment

### 2.1. Materials

Generally, the chemicals employed in this study were of the best analytical grade available, obtained from Sigma-Aldrich Chemical Co (St. Louis, MO, USA) or Merck Ltd. (Kenilworth, NJ, USA), and used without further purification. All of the solvents used for spectral measurements were of spectrometric grade. Ethylene glycol (EG, was obtained from EG J. T. Baker, (Radnor, PA 19087, USA); polyvinylpyrrolidone M.W. 44000 (PVP, was obtained from Alfa Aesar, (Heysham, LA3 2XY, England); silver nitrate (AgNO3 99%, was obtained from Mallinckrodt Chemicals, (Blanchardstown, Dublin 15, Ireland); cupric chloride (CuCl2, 99% was obtained from Merck Ltd. (Kenilworth, NJ, USA); and rhodamine 6G (R6G, was obtained from Sigma-Aldrich Chemical Co (St. Louis, MO, USA). All reagents were used without further purification in our experiments. The pesticides chlorpyrifos (CAS2921-88-2); thiram (CAS137-26-8); carbaryl (CAS63-25-2); and paraquat (CAS75365-73-0) were all purchased from Sigma-Aldrich Chemical Co. (St. Louis, MO, USA), Fipronil (CAS 120068-37-3) was purchased from TCI (KITA-KU, Tokyo, Japan).

### 2.2. Apparatus

Absorption spectra were generated using a Thermo Genesys 6 UV-visible spectrophotometer(Waltham, MA, USA), and fluorescence spectra were recorded using a HORIBA JOBINYVON Fluoromax-4 spectrofluorometer (Minami-ku Kyoto, Japan)with a 1 nm bandpass filter in a 1 cm cell length at room temperature. AFM images of the nanostructures were obtained using a NanoMagnetics Instrument Ltd. ezAFM. (Summertown, Oxford, UK) TEM images of the nanostructures were taken using a JEOL JEM-2100F J microscope (Musashino Akishima, Japan)at an accelerating voltage of 100 kV. An aqueous solution containing the compound was deposited onto a carbon-coated copper grid. Dynamic light scattering (DLS) measurement was recorded using a SZ-100—HORIBA(Minami-ku Kyoto, Japan). The fluorescence images were taken using Leica AF6000 fluorescence microscopy(Leitz-Park Wetzlar, Germany) with a DFC310 FX Digital color camera through related cubes. In this study, the UV light cube (in which the light passed through a 390/10 nm band pass filter and the emission was collected through a 410 nm long pass filter), blue light cube (in which light passed through a 470/20 nm band pass filter, and the emission was collected through a 510 nm long pass filter), and green light cube (in which light passed through a 520/20 nm band pass filter, and the emission was collected through a 590 nm long pass filter) were used to collect the fluorescence images of pesticides and R6G.

### 2.3. Synthesis of Ag Nanowires (Ag NWs)

The synthesis of Ag nanowires was achieved according to a procedure reported elsewhere [28]. A typical synthesis involves ethylene glycol (EG) as both the solvent and the reducing agent, with AgNO3 and poly(vinvlpyrrolidone) (PVP, MW = 40,000) as the Ag precursor and the polymeric capping agent, respectively. In this synthesis, the CuCl_2_ species can be added to facilitate the anisotropic growth of Ag nanowires. In a typical synthesis, 20 mL of EG was added to a disposable glass vial containing a Teflon stirrer bar; the vial was then suspended in an oil bath (temperature = 150 °C) and heated for 1 h under magnetic stirring (400 rpm). At 1 h, 160 mL of a 4 mM CuCl_2_ solution in EG was injected into the heated EG. The solution was then heated for an additional 15 min. Next, 6 mL of a 0.147 M PVP solution in EG (concentration calculated in terms of the repeating unit) was injected into the heated EG, followed by the addition of 6 mL of a 0.094 M AgNO_3_ solution in EG. The color of the reaction solution changed as follows: initially clear and colorless to yellow (within 1 min), to red-orange (within 3 min), to green, beginning to become cloudy (within 5 min), to cloudy, with a gradual shift from green to brown-red (within 30 min), and finally to an opaque gray with wispiness, indicating the formation of long nanowires (within 1 to 1.5 h). Upon the formation of Ag nanowires, the reaction was quenched by cooling the reaction vial in a room temperature water bath. The solution was centrifuged at 10,000 rpm for 10 min to ensure the complete collection of products, then washed with double-distilled water and centrifuged (10,000 rpm,10 min), and then washed with ethanol and centrifuged three times (6000 rpm, 15 min) to remove the EG and PVP on the surface of the products. The final products were preserved in ethanol for further characterization.

### 2.4. Construction of a 3D Nanowire Chip and Measurement

We prepared a solid thin film for which the 3D random crossed-wire woodpile (RCW) nanostructures and sample preparation are shown schematically in Figure 1. A circular well on a glass plate with a diameter of 0.5 cm and depth of 0.02 cm was used, onto which we sprinkled 20 µL of the Ag NW-containing solution (the concentration was about 2.6 optical density (OD), as shown in Figure 2a) and allowed it to dry; this step was repeated several times to create a 3D nanowire disarray network. In this way, we obtained 20 µL × 1, 20 µL × 2 … 20 µL × N of the 3D-RCW chips for drop casting of the analyte-containing aqueous solution and drying, ready for SERS and PEF detection.

### 2.5. Raman Measurement

Raman micro-spectroscopy measurements were performed on the Micro Raman Identify Dual system (MRID-Raman, ProTrusTech Co., Ltd., Tainan, Taiwan) mounted with one TE cooled CCD of 1024 × 256 pixels as integrated by Protrustech Corporation Limited. The system with a 50 × long working distance lens (Olympus America Inc., New Hyde Park, NY, USA) was operated at an excitation wavelength of 532 nm, with ~1 mW power, in order to avoid laser-induced degradation. Raman spectra were recorded at a spectral resolution of 1 cm^−1^ in the spectral range between 400 and 2500 cm^−1^. The exposure time for Raman spectra was 1 s and each spectrum was accumulated for one time. The accumulation time and the laser power were the same for all Raman spectra in the case of no special instructions. The measurement method was as follows: We dropped the analyte aqueous solution with certain concentrations (20 µL at a time) into the well of the 3D-RCW nanochip, as shown in Figure 1, and then dried it in a dry bath incubator (40 °C) for measurement. To measure the data reproducibility and repeatability of 3D-RCW nanochip, SERS spectra was collected from seven different places on each chip and then averaged, as the standard spectrum. The algorithm of data deviation comes from the intensity-subtraction between every spectrum and standard spectrum. The stability of the 3D-RCW nanochip was shown by the SERS signals still being detectable and the intensity less than 10 percentage after a chip was placed in the atmosphere for several days.

## 3. Result and Discussion

### 3.1. Characterization of Ag NWs

The silver nanowires were synthesized using a solution-based polyol process [28]. Figure 2a shows the absorption spectra for Ag NWs in double-distilled water. The dominant surface plasmon resonance (SPR) peaks for the silver nanostructures in solution were observed to be consistent with the typical optical properties of silver nanoparticles and nanowires synthesized via the polyol process [12]. Here, we focus on the nanowires and assign the SPR peak at 400 nm to the transverse SPR mode (LSPR) of the Ag NWs [13], and the broad absorption covers the wavelength range for the visible range of emission wavelengths for most commercially available fluorophores, which is suitable for PEF applications. In addition to the observations described above, the absorption spectrum of Ag NWs in aqueous solution displayed a broadening that can be attributed to the coupling of the SPR due to the decrease in the spacing between the nanowires. The inset of Figure 2a shows a real-color photograph of Ag NWs in solution with a cloudy yellow-green color (a typical color of silver colloids) and a dominant transverse SPR peak occurring at approximately 400 nm. Figure 2 also presents the microscopy, AFM, and SEM images showing the average values for the diameter and length of the Ag NWs. Silver nanowires were characterized by optical microscopy, AFM, and SEM after the synthesis, as shown in Figure 2b–d. Although there is a broad dispersion in size, characteristic values of (146.7 ± 12) nm in diameter and (55.3 ± 8) μm in length were collected by dynamic light scattering (DLS). We extracted the size of NWs from statistical analysis of several SEM images and found that they were 80–120 nm in width and >10 μm in length. The average aspect ratio (length/diameter) of the nanowires was more than 100. The thickness of the PVP coating on the surface of the nanowires was measured to be 20–25 nm.

### 3.2. Construction of an Ag NW 3D Nanostructure

The schematic in Figure 1 shows the procedure used for fabricating Ag NW-constructed 3D random crossed-wire woodpile (3D-RCW) nanostructures through a very conventional sprinkling method. First, 20 µL of 2.6 O.D. Ag NW solution was drop cast onto a 0.5 cm × 0.2 mm well on a glass slide, and then the solvent was evaporated in vacuum, which corresponded to one cycle of spreading. In this way, multistacked 3D-RCW nanostructures with various numbers of layers were fabricated and the layer and density of crossed-wire 3D-RCWs were spread-cycle dependent. Eventually a 3D nanostructure dry chip was prepared and analytes could be placed onto the well for collection of the plasmon-enhanced spectra. To establish the optimal plasmon enhancement, we examined the SERS characteristics for the 3D-RCW chip based on rhodamine 6G (R6G). Figure 3a shows a comparison of the SERS intensity of R6G obtained from variable densities of 3D-RCW chips. Here, the intensity of the Raman spectra for a constant concentration of R6G (10 μM × 20 µL) apparently increased up to the third cycle of Ag NW sprinkling (20 µL × 3). The high SERS signal enhancement can be explained by the fact that with more cycles of spreading, there are more layer formations, more cross-stacked nanowires, and then more z-direction hot spots in the 3D-RCW, eventually inducing strong plasmonic coupling along the vertical direction [29,30]. On the other hand, the signal intensity decreased with more than three cycles of spreading, which can be explained by the fact that the penetration depth decreases with higher densities of 3D nanostructures regardless of the laser source or analyte SERS signal. The transmittance of 3D-RCW decreases dramatically as the number of stacking layers increases, which means the incident light intensity is impeded. Accordingly, it is more difficult for CCD to collect the SERS signals of analytes buried in the deeper-layers. Meanwhile, if the molecules are distributed in more layers, the number of molecules in each layer will decrease. Nevertheless, transmittance is the criterion to build up an optimized 3D-RCW for SERS measurement. Figure 3b presents data for the reproducibility, repeatability, and stability of 3D-RCW for R6G SERS signals. The data error distribution was between 10~15% at several points of measurement, and these signals could still be detected when using a chip that was placed in the atmosphere for over 80 days. Finally, the minimum detection limit for R6G was determined to be 10^−11^ M in 20 μL volumes using a 3D-RCW (20 × 3) chip to measure the SERS signal of R6G (Figure 3c).

### 3.3. Plasmon-Enhanced Fluorescence

A high enhancement in fluorescence emission, improved fluorophore photostability, and a significant reduction in the fluorescence lifetimes were obtained using a high density of Ag NWs. These quantities depend on the surface loading of Ag NWs on a glass slide, where the enhanced fluorescence emission increases with the density of Ag NWs. Thus, we also checked the 3D-RCW for PEF for R6G. Figure 4a shows fluorescence microscopy images for Ag NW-based plasmonic-emission enhancement for R6G. It is clear that the area covered by Ag NWs reveals apparent fluorescence emission that is much higher than that without Ag NWs. By using a similar investigation method, Figure 4b–f show a comparison of the fluorescence images for PEF of R6G obtained for variable densities of 3D-RCW; a comparison of the spectra and emission intensities are shown in Figure 4g. The emission enhancement initially increases rapidly before flattening out, and is proportional to the variable densities of the Ag NW network platform. That is, for higher density coatings of 3D-RCW, the PEF effect vanishes due to a similar reason to that given above. However, the decreased signal is not so critical due to fluorescence offering a larger cross section than that of Raman. Nevertheless, we found that the PEF of 3D-RCW to R6G increased by more than 1000-fold compared to that of the free R6G in the solid state.

The results in Figure 3 and Figure 4 confirm the strong SERS and PEF dual effects in our 3D-RCW chip. The intensified LSPR arises due to strong electromagnetic field enhancement that occurs at the abundant formation of hot spots, which is ideal for LSPR-based SERS and PEF mechanisms [21,31]. Figure 5 clearly shows that more apparent fluorescence highlights occur at the intersection and ends of the nanowires, and we observed that the aspect ratio of the NWs controls the occurrence of hotspots and/or antennas. The bigger aspect ratios of the nanoneedles present more intersections to reveal more hot spots with less antenna effects because of difficult conduction (Figure 5a), while shorter nanorods present more bright spots at both ends with fewer crossover points (Figure 5b). That is why we did not observe a similar result in the nanoparticle system. Nevertheless, this is the first time that hot spot formation on a nanoplasmonic sensor has been unequivocally confirmed by utilizing PEF imaging.

### 3.4. Appearance of 3D-RCW

As shown in Figure 1, the formation of a 3D-RCW composed of 3D multilayered stacks of Ag NWs can provide both in-plane (xy-plane) and out-of-plane (z-direction) plasmonic coupling effects for both parallel or cross nanowire stacking. There are some studies that mention 3D nanowire stacking and determine that cross stacking shows the best simulated electric field enhancement. Thus, these kind of 3D Ag NW woodpile structures offer orthogonal nanowires piled up along the z-direction [23,30,32,33]. That is, the xy-plane hot spot combines with the z-direction hot spot to generate a 3D hotspot region, and the maximum electric field enhancement variations should be stacking-layer dependent. In a regular hot spot array constructed from 3D nanowires, the maximum E-field enhancement increases from 2 to 5~6 layers of stacking, and plasmonic nanostructures exhibit a redshifted LSPR peak with increasing number of stacking layers. Thus, a ca. 200 nm thick 3D hotspot network is optimal [30]. In our case, it is reasonable that three cycles of Ag NW spreading is thick enough to permit measurement of the optical properties and SERS performance as well as for the penetration depth limitation for visible light.

With a regular 3D nanowire film, as mentioned in the above reference, there is always a defect (gap, interstice) effect. In other words, different minimum mesh sizes around the nanowires must be constructed to adjust the distance of defect and fit different sized analytes. Furthermore, one must consider the distance and kinetic equilibrium problem between molecules and metal in solution-state detection. That is why most chip designs cannot be used for both SERS and PEF. We used the 3D-RCW chip to analyze a molecule by drop casting an aqueous solution and allowing it to evaporate. Under this condition, one ensures that every molecule is absorbed on the surface of the metal. There are no solubility and solvent problems, so we can collect the water soluble and insoluble molecules. Consequently, for the random mesh interval characteristics of 3D-RCW, the same molecule may present SERS (remain on the surface of a nanowire) and PEF (short distance to another nanowire). Nevertheless, an abundantly and evenly distributed 3D hot spot network is the impact factor. The SERS spectra obtained for R6G on a freshly prepared 3D-RCW composite film and that stored for 80 days (Figure 3b) without vacuum sealer storage revealed that the SERS intensity for R6G from the stored substrate was reduced by less than 10%, indicating that the 3D-RCW composite film was quite stable when stored over a period of time. It is concluded that 3D-RCW has the following superiorities over regular 3D nanowire film: (1) Solid-state detection: can detect many kinds (size, solubility) of molecules; (2) Tiny testing volume: a small-area well for analytes to fill ensures that the molecules can spread more uniformly; (3) Both SERS and PEF can be measured by using the same chip; (4) Low cost chip fabrication: no antibody, no labeling, electrical-etching technology.

### 3.5. Application of 3D-RCW

To further investigate the practical applications of the 3D-RCW nanochip, a test for organic pollutant pesticides relevant to environmental monitoring was performed. Thiram, a dithiocarbamate compound, is widely used as a fungicide in agriculture and a bactericide in medical treatment [34]. The SERS spectra for different concentrations of thiram dispersed onto a 3D film were tested, as shown in Figure 6a. In comparison with the normal Raman spectrum of thiram, the vibrational peaks due to thiram appeared at 567 cm^−1^, 1150 cm^−1^, 1386 cm^−1^, and 1516 cm^−1^, corresponding to the C–S stretching, N=C=S stretching, C–N stretching, and C–H wagging modes, respectively [35]. The insert shows the available curve of the peak (1386 cm^−1^) intensity with respect to linear detecting amounts ranged from 0.1 to 10 μM, with a limit of detection (LOD) of ~0.1 μM (~0.02 mg/Kg), which is much lower than the maximal residue limit (MRL) of 7 mg/Kg in fruit prescribed by the U.S. Environmental Protection Agency (EPA). Here, the LOD was defined as the lowest quantity of analyte that can be detected in our system. For example, the point with the smallest number in the insert of Figure 6. Figure 6b shows the SERS spectrum for carbaryl, which is a carbamate pesticide that is banned in many countries. For example, the MRL (maximum residue limits) for carbaryl in apples is 1 mg/Kg (GB2763-2012.). Peak features at 1382 cm^−1^, 1442 cm^–1^, and 1578 cm^–1^ can be obviously observed in the Raman spectra but changes in both relative intensities and the position of the bands were observed [36,37]. A strong peak observed at 1382 cm^–1^ is due to the symmetric vibration of the naphthalene ring. The peak at 1442 cm^–1^ arises from C–H wagging modes of the monosubstituted naphthalene ring. The strong peak at 1578 cm^–1^ can be assigned to the stretching of C=C double bonds in the naphthalene ring. The insert shows the available curve of the peak (1382 cm^−1^) intensity with respect to linear detecting amounts range from 5.0 to 100 μM, the LOD value was found to be ~5.0 μM (1.00 mg/Kg). The assignments of the Raman modes of the pesticides we used in this manuscript are listed in Table 1, taken from references as indicated in the manuscript.

Among the pesticides, paraquat with moderate toxicity, is widely used in agricultural practices, and its permissible residue for apples and pears is regulated to be lower than 0.05 mg/Kg in the USA, China, and most other countries [38]. Figure 6c shows the characteristic paraquat SERS peaks at 837 cm^−1^, 1191 cm^−1^, 1293 cm^−1^, and 1642 cm^−1^, which is attributed to the C–N stretching mode, C=C bending vibration mode, C-C structural distortion mode, and C=N stretching mode, respectively [39]. The insert shows the SERS peak at 1642 cm^−1^, corresponding to the linear detection amounts, which ranged from 0.1 to 50 μM, the LOD value was found to be ~0.1 μM (0.02 mg/Kg). We showed how 3D-RCW can be used to detect fipronil, which is rarely detected with an effective quick screen. Fipronil exhibits high sensitivity to insects that are resistant to cyclopentadiene, organic phosphorus, organic chlorine, pyrethroids, carbamate pesticides, and those that have no cross-resistance to existing pesticides [40]. Recently, fipronil sulfone has been detected in eggs at much higher levels than the maximum residue limit. The European Food Safety Authority (EFSA) has set a more stringent limit of 0.005 mg/Kg in poultry muscle and eggs [41]. From the spectra shown in Figure 6d, we can see the strongest characteristic peak occurs at approximately 2253 cm^−1^, which is likely due to the nitrile (−C≣N) group [42] that is unique and can be differentiated from many other analytes. The insert shows that the Raman signal intensity at 2253 cm^−1^ was positively correlated with the amount of fipronil. The amount ranged from 5.0 to 200 μM when for linearity and the LOD value was found to be 5.0 μM (2.18 mg/Kg). As we know, it is difficult to detect fipronil using Raman spectroscopy, which is mainly due to the weak interaction between fipronil and metal and the low solubility of fipronil that can easily crystallize-out of aqueous solutions. In our study, microliter levels of analyte solutions were used followed by drying; the solid-state detection platform and tiny portion demonstrate the better performance and higher sensitivity of the 3D-RCW.

Finally, to demonstrate the proof-of-principle use of 3D-RCW in PEF-based applications, pesticides (thiram, carbaryl, paraquat, and fipronil in Figure 6) were drop cast onto chips (coated with Ag NWs with 20 µL × 3) and measured using a fluorescence spectrometer and fluorescence spectroscopy. Based on the observation and discussion above, we used the same chip in Figure 6 to make sure that systemic emission data did not drop at a high concentration of nanowires. Figure 7 shows the real-color photographs of the fluorescence emission imaging from the 3D-RCW chips taken through an emission filter and the related measured fluorescence emission spectra for the pesticides. There were clear blue, cyan, and blue-green colors of fluorescent microscopy images revealed in our 3D-RCW chips for carbaryl, paraquat, and fipronil, respectively. Spectra measurements in the inserts of Figure 7 indicate the nonlinear PEF enhancements with increasing amounts of pesticides, thus, we estimated the LODs for carbaryl (50 μM, 10.00 mg/Kg), paraquat (5 μM, 1.28 mg/Kg), and fipronil (20 μM, 8.74 mg/Kg) based on the lowest concentration (the smallest amount) of pesticide, which was non-fluorescent in the control chip (without nanowire), that could be detected by using fluoresce microscopy. This is the first time that pesticides were detected using fluorescent images. Since carbaryl and fipronil pesticides are difficult to detect by Raman or SERS spectra assays, PEF provides an opportunity to detect pesticides, with similar or better LOD than SERS, with fluorescence signals.

We summarize some previous reports in Table 2 and compare the SERS substrates and analytes used. In order to achieve high detection limits, most of these researchers fabricated their chips with rare materials, complicated processes, or more than one chemical reaction, which means many of the chips were expensive to produce. The benefits of our chip compared to those listed in Table 2 include:Most used Au, while we used a very stable Ag.Most used nanoparticles, while we used nanowires.A 3D substrate was constructed in our chip.Our chip and analytes were prepared simply.Most detected just one or two pesticides, while we detected four pesticides that belong to three types of pesticides (carbamate, paraquat, and fipronil).Fipronil is hard to detect by SERS, but our chip detected it.We detected fluorescence and Raman using the same chip.

### 3.6. Further Applications

Ag NWs can be impregnated within or assembled onto solid, flexible substrates using filter paper, fiber mats, elastomers, and plastics, producing 3D flexible plasmonic substrates. The flexible plasmonic substrates display outstanding attachment properties to curved surfaces. Direct attachment of flexible SERS substrates to human skin can also enable in vitro detection of biochemicals or biomarkers from perspiration.

3D-RCW can be used to enhance emission intensity for a fluorophore sensor to improve the detection limit.

The combination of a portable spectrometer with a low-cost but highly sensitive and flexible plasmonic substrate should be commercialized for on-site chemical analysis in environmental monitoring, food safety, forensic science, and point-of-care medical diagnostics.

## 4. Conclusions

This article has emphasized the utility of 3D stacked Ag NWs for enhancing plasmonic coupling effects. Plasmon-enhanced fluorescence (PEF) and surface-enhanced Raman scattering (SERS) spectra data can be collected by using a nanochip with a 3D-RCW platform. Based on our study, a 3D-RCW nanostructure can provide rich antenna and hot spot effects, and optimization of the PEF or SERS effect in terms of the detection limit was explored based on layer-by-layer construction between silver nanowires. We successfully observed PEF and SERS effects for R6G and pesticides with this platform, which can be used to build a novel dual analysis and extend the range of fluorescence biosensors. This 3D nanoplatform shows promising potential as a cheap, robust, and portable sensing platform for future applications.

## Figures and Tables

**Figure 1 molecules-26-00281-f001:**
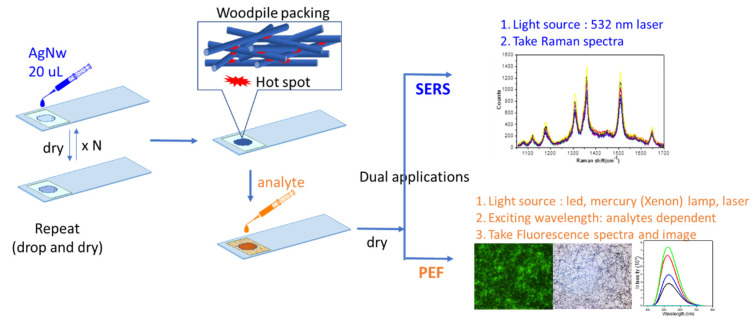
Construction of a 3D random crossed-wire woodpile (RCW) and preparation for analyte detection for surface-enhanced Raman scattering (SERS) and plasmon-enhanced fluorescence (PEF), respectively. The red color region in the woodpile graph indicates the hot spot.

**Figure 2 molecules-26-00281-f002:**
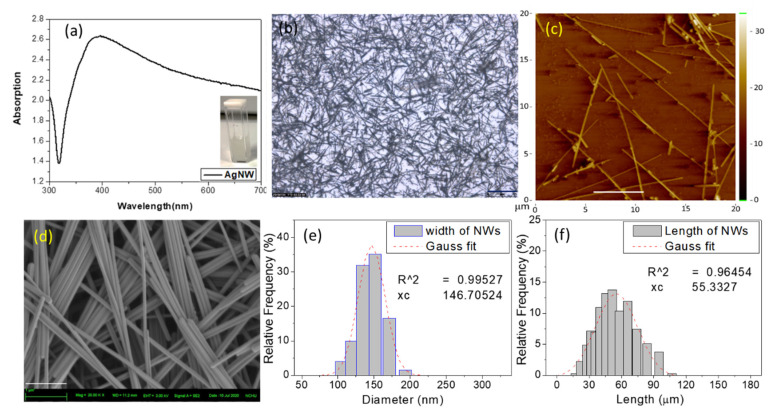
(**a**) Absorption spectra for Ag nanowires (NWs) (5 mg/mL); the insert shows the real-color aqueous solution photograph. (**b**) Microscopy, scale bar is 25 μm; (**c**) AFM, scale bar is 5 μm; (**d**) SEM image of Ag NWs, scale bar is 1 μm. (**e**) Width and (**f**) length data from dynamic light scattering (DLS).

**Figure 3 molecules-26-00281-f003:**
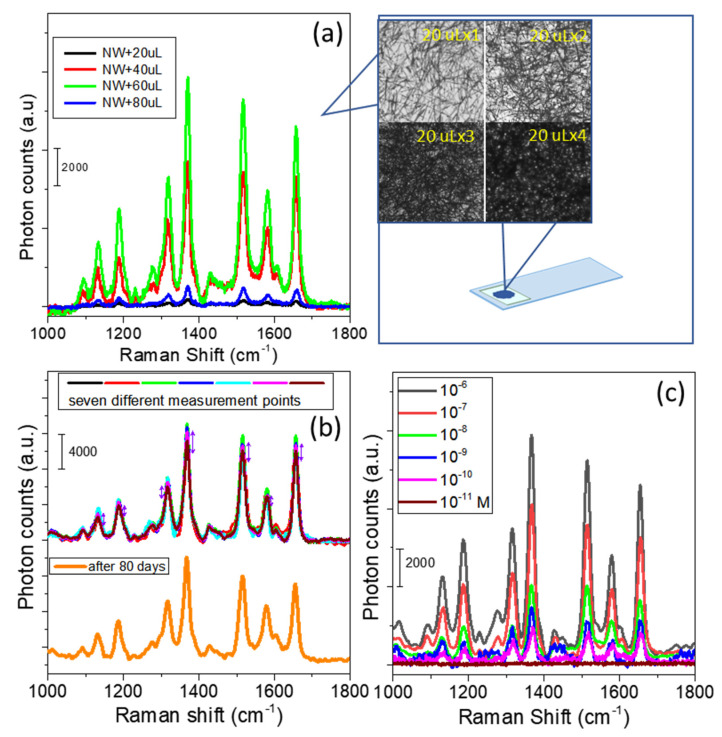
(**a**) 20 μL of 10 μM R6G sprinkled onto glass was coated with variable densities of Ag NWs (20 μL × 1, 2, 3, 4, 5). (**b**) SERS collection at seven different locations on a freshly prepared 3D-RCW chip (top); these signals dropped by less than 10% when using a 3D-RCW prepared 80 days previously and left in the atmosphere. (**c**) SERS detection limit for R6G using a 3D-RCW chip with a 20 × 3 Ag NWs coating density. In (**a**), 10 uL of analyte was used to detect the SERS spectra.

**Figure 4 molecules-26-00281-f004:**
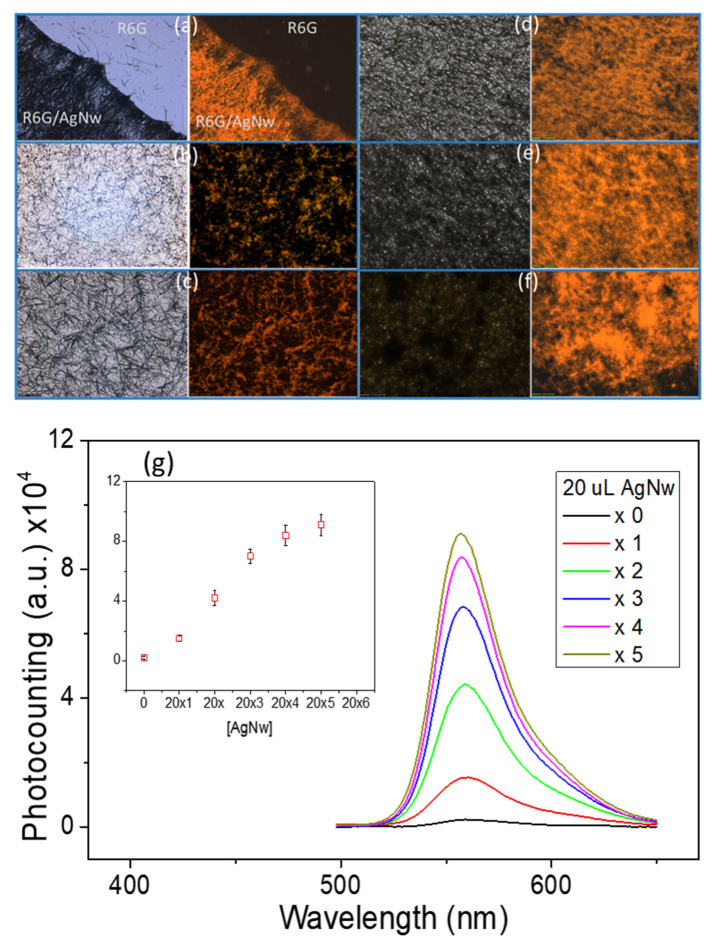
(**a**–**f**) Bright-field (left) and fluorescent (right) images, the blue-light cube was used for detection of fluorescence microscopy (as described in the Section 2). (**a**) Proof for PEF: R6G is spread onto a glass slide with one half coated with Ag NWs and the other half with no coating. (**b**–**f**) 20 μL of 10 μM R6G was sprinkled onto glass and coated with variable densities of Ag NWs (20 μL × 1, 2, 3, 4, 5; as described in the Section 2). (**g**) Quantitative emission intensities for (**b**–**f**) from the fluorescence spectra (Ex: 480 nm). The inset shows a plot of the emission intensities at 560 nm.

**Figure 5 molecules-26-00281-f005:**
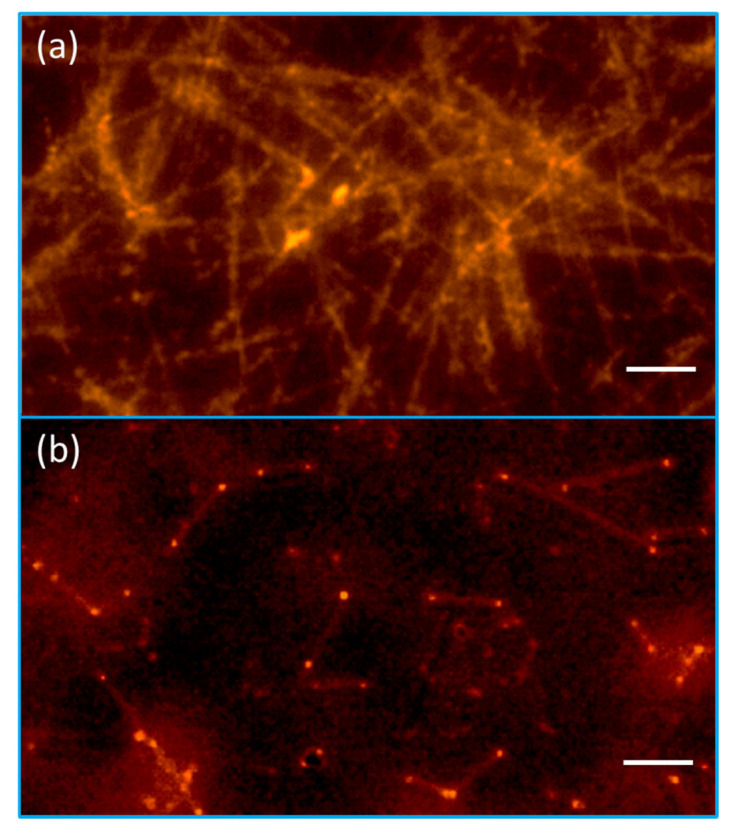
Hot spot distribution for 3D-RCW at the intersection of nanowires (**a**) and at both ends of the nanorods (**b**). The green-light cube was used for this detection of fluorescence microscopy. Scale bar = 10 μm.

**Figure 6 molecules-26-00281-f006:**
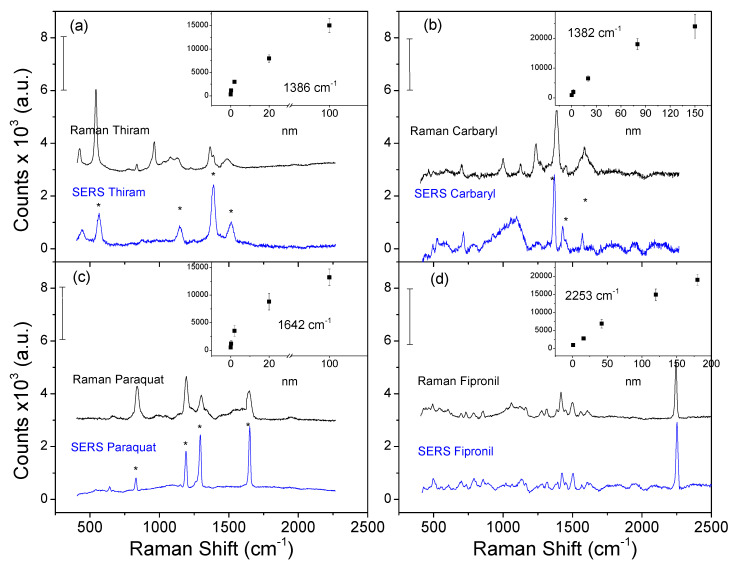
The application of a 3D-RCW chip for direct and rapid detection of pesticides (**a**) thiram, (**b**) carbaryl, (**c**) paraquat, and (**d**) fipronil with rational detecting ranges (nanomole), as plotted in the inserts. * improve the discriminant accuracy.

**Figure 7 molecules-26-00281-f007:**
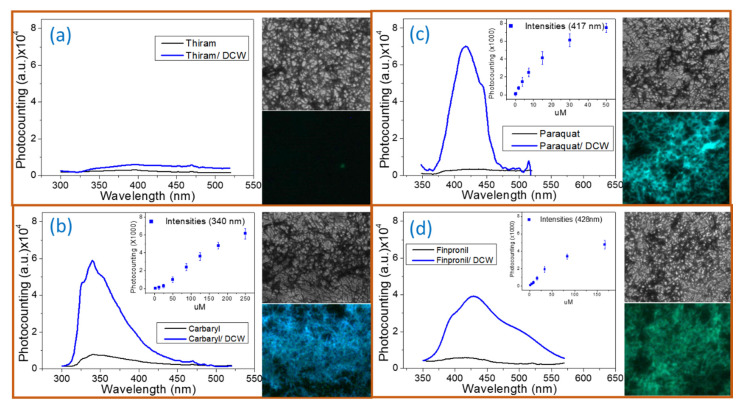
Emission spectra, bright-field (top) and fluorescent images (bottom), for (**a**) thiram, (**b**) carbaryl, (**c**) paraquat, and (**d**) fipronil; the UV-light cube was used for detection of fluorescence microscopy. Inserts in the spectra figures present the nonlinear PEF effects with increasing portions of pesticides.

**Table 1 molecules-26-00281-t001:** Assignments of the Raman modes in pesticides, taken from references as indicated in manuscript. SERS unit (cm^−1^).

Thiram		Carbaryl		Paraquat		Fipronil	
SERS	Mode	SERS	Mode	SERS	Mode	SERS	Mode
567	C–S stretching			837	C–N stretching		
1150	N=C=S stretching			1191	C=C bending		
1386	C–N stretching	1382	symmetric vibration (naphthalene ring)	1293	C–C structural distortion		
1516	C–H wagging	1442	C–H wagging of naphthalene ring				
		1578	stretching of C=C double	1642	C=N stretching		
						2253	nitrile (−C≣N)

**Table 2 molecules-26-00281-t002:** Comparison of substrates and limits of detection (LODs) of various methods from the literature. * improve the discriminant accuracy.

Analytes	LODs	Substrates	Nobel Metal Nanostructure (Exciting Laser)	Refs
rhodamine 6G (R6G)	-	SERS signal: nanostars > nanotriangles > nanospheres	Gold nanostructures (785 nm lasering)	[43]
cyclotrimethylenetrinitramine (RDX)	0.15 mg/L	glass slide	Au NP suspension (785 nm lasering)	[44]
methamphetamine	-	flow-focusing microfluidic (PDMS)	controlled Ag-NP/salts aggregation (633 nm lasering)	[45]
DNA bases 4-aminothiophenol (4-ATP)	1 fM- 10 aM	slippery liquid-infused porous surface-enhanced Raman scattering (SLIPSERS)	Au NPs + analyte (633 nm lasering)	[46]
polychlorinated biphenyls (PCBs)	1PPb			
naphthalene	25 ppm	“chemical tether” to anchor NPs on a quartz substrate	Au NPs (633 nm lasering)	[47]
2,4,6-trinitrotoluene (TNT), 2,4-dinitrotoluene (2,4-DNT), 1,3,5-trinitrobenzene (TNB)	0.89~ 94 pg.	clusters of NPs on the cellulose fibers of the paper.	Au NPs (785 nm lasering)	[48]
melamine	1 ppb	physical vapor deposition and then electrochemical deposition	(formation of Ag or Au metal film over nanosphere (FON) surface) (785 nm lasering)	[49]
parathion-methyl thiram chlorpyrifos	2.60 0.24 3.51 ng/cm^2^	nanoparticle solution was dropped uniformly on the sticky side of adhesive tape	Au NPs (paste and peel off’ sampling approach.) (633 nm lasering)	[50]
crystal violet mitoxantrone	10 nM 1 nM	electron beam lithography and nanotransfer printing	Au nanopattern (633 nm lasering)	[51]
melamine	33 ppb	lithographically, substrate is composed of an array of pyramidal-shaped pits etched into silicon.	Ag/Au NPs pyramidal Klarite^@^ substrates	[52]
organophosphate malathion, heroin, cocaine	413 pg, 9 ng, 15 ng	inkjet-printed paper-based dipsticks and swabs	Au nanoclusters (785 nm lasering)	[53]
perchlorate	0.343 ± 0.025 mg/L	electron beam lithography	Au ellipse dimer array (785 nm lasering)	[54]
uranyl solution	120 nM.	chemical modification on nanostructure surface	Au nanostars. (785 nm lasering)	[55]
cyanide	173 ppt	lab-on-a-bubble (LoB) assay	Au NP-coated LoBs	[56]
hexavalent chromium	59 ppb	a capture matrix with self-assembled monolayer on immobilizing nanocluster	Au/MEPH + substrates (785 nm lasering)	[57]
paraoxon fenitrothion	10^−12^ M	metal-organic framework (MOF-5)	Au-grating (785 nm lasering)	[58]
methyl parathion, edifenphos, ethyl paraxon	*	spectra collection, then chemometric methods (standard normal variate variance (SNV) method)	Ag sol gel (532 nm lasering)	[59]

## Data Availability

The data presented in this study are available in this article.

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
