# Peer review of "A 3D Plasmonic Crossed-Wire Nanostructure for Surface-Enhanced Raman Scattering and Plasmon-Enhanced Fluorescence Detection"

_molecules, 2021, doi:10.3390/molecules26020281_

Round 1

Reviewer 1 Report

I have reviewed the manuscript and I believe that with the modifications made to the manuscript it could be accepted for publication in the journal Molecules.

Author Response

Response to Reviewer 1 Comments

Comments and Suggestions for Authors

I have reviewed the manuscript and I believe that with the modifications made to the manuscript it could be accepted for publication in the journal Molecules.

Response:

We thank the reviewer for the affirmation.

Reviewer 2 Report

The paper by Fuh-Jyh Jan et.al. “A 3D plasmonic nanostructure for surface-enhanced Raman scattering and plasmon-enhanced fluorescence detections” describes the design, fabrication, and proof-of-principle measurements of cross-wired nanostructure with dual capability for SERS and PEF. The paper needs major revisions as there is a large list of misprints (see examples below) and English needs extensive editing as it was very hard at times to understand what the authors meant. Given the novelty of the study, I would recommend reevaluating its scientific soundness in the second round of review after the authors address the English editing issue.

I would also recommend changing the title of the article: to specify “nanowire” for example “A 3D plasmonic crossed-wire nanostructure …” and correct “detections” to “detection”.

Additionally, the following is an incomplete list of misprints:

Page 2 line 58: “is am increase” should be “is an increase”

Page 2, line 89: “serval orders of magnitude” should be “several orders of magnitude”

Page 4 line 149: “The exposure time was 1 s with 1 accumulation”. What does it mean? Is it “One spectrum was acquired with 1 sec acquisition time”? Some methodological details are not clearly described.

Page 4 line 151-154: Here I could only guess about what the procedure was.

Figure 1: many misprints give an impression of a very sloppy job.

Figure 2: AFM, scale bar is stated to be 5 μm. It is not very visible where the scale bar is and how long it is.

Figure 4: : ex, 470/20 nm; em, 510 nm lp filter is way too technical for a Figure caption. These details need to be moved to “methods” section. lp needs to be spelled out: “long pass” for optical filters as lp is also used for frequency filters as “low pass”.  

Page 5: The authors use “analyst” throughout the text. For example, page 5 line 205 “analyst SERS signal”. I am guessing they mean “analyte“ as in “SERS signal of an analyte”.

Line 303 “in the insert of Figure” – what is figure #.

The authors discuss the permissible levels of paraquat in mg/kg but report LOD in μM, how do these compare?

Lines 353-355: “based on that these pesticides can fluorescence imaging or not with respect to the nonfluorescent control contents in the solid-state”. I don’t understand the meaning of this statement.

Author Response

Response to Reviewer 2 Comments

Comments and Suggestions for Authors

The paper by Fuh-Jyh Jan et.al. “A 3D plasmonic nanostructure for surface-enhanced Raman scattering and plasmon-enhanced fluorescence detections” describes the design, fabrication, and proof-of-principle measurements of cross-wired nanostructure with dual capability for SERS and PEF. The paper needs major revisions as there is a large list of misprints (see examples below) and English needs extensive editing as it was very hard at times to understand what the authors meant. Given the novelty of the study, I would recommend reevaluating its scientific soundness in the second round of review after the authors address the English editing issue.

I would also recommend changing the title of the article: to specify “nanowire” for example “A 3D plasmonic crossed-wire nanostructure …” and correct “detections” to “detection”.

Response:

Thank the reviewer for the comment. The Title of manuscript has been rewritten as: A 3D plasmonic crossed-wire nanostructure for surface-enhanced Raman scattering and plasmon-enhanced fluorescence detection.    

Additionally, the following is an incomplete list of misprints:

Page 2 line 58: “is am increase” should be “is an increase”

Response:

Thank the reviewer for the correction. We have revised the wrong typing.--------in line 58.

Page 2, line 89: “serval orders of magnitude” should be “several orders of magnitude”

Response:

Thank the reviewer for the correction. We have revised the wrong typing.----------in line 89.

Page 4 line 149: “The exposure time was 1 s with 1 accumulation”. What does it mean? Is it “One spectrum was acquired with 1 sec acquisition time”? Some methodological details are not clearly described.

Response:

Thank the reviewer for the comment. Normally, for the description of spectra quality as: “The exposure time for Raman spectra with low SNR (signalto-noise ratio) was 50 ms and each spectrum was accumulated for one time, while the exposure time for Raman spectra with high SNR was 10 s and each spectrum was accumulated for 30 times.” (Optics Express 2014, 22, 12102-12114)

In our case, this sentence “The exposure time was 1 s with 1 accumulation.” means “The exposure time for Raman spectra was 1 s and each spectrum was accumulated for one time.” However, combine with the next query from Reviewer, we rewrite this section line 156-162 as :

“Raman spectra were recorded at a spectral resolution of 1 cm−1 in the spectral range between 400 and 2500 cm−1. The exposure time for Raman spectra was 1 s and each spectrum was accumulated for one time. The accumulation time and the laser power were the same for all Raman spectra in the case of no special instructions. The measurement method: dropped the analyte solution with certain concentration (20 µL at a time) into the well of the 3D RCW nanochip, as shown in Figure 1, and then dried in a dry bath incubator (40 ℃) for measurement.”

Page 4 line 151-154: Here I could only guess about what the procedure was.

Response:

Thank the reviewer for the correction. We are sorry for the misunderstanding caused by the unclear description. We removed some incorrect and inappropriate words or sentences and rewritten this section line 156-162, as last response shown.

Figure 1: many misprints give an impression of a very sloppy job.

Response:

Thank the reviewer for the comment. We replotted the Figure 1, which described the packing/stacking modus of nanowire and detection method of analysts. Illustration of 2.4 and 2.5 in Experiment

Figure 2: AFM, scale bar is stated to be 5 μm. It is not very visible where the scale bar is and how long it is.

Response:

Thank the reviewer for the comment. We replotted the scale in AFM diagram in Figure 2, with same size but thicker.

Figure 4: : ex, 470/20 nm; em, 510 nm lp filter is way too technical for a Figure caption. These details need to be moved to “methods” section. lp needs to be spelled out: “long pass” for optical filters as lp is also used for frequency filters as “low pass”. 

Response:

Thank the reviewer for the comment. We described the detail definitions for UV-cube (ex: 390/10 bp, em 410 lp) and blue cube (ex: 470/20 bp, em 510 lp) in 2.2 section of 2. Experiment. Figure captions of Figure 4, 5 and 7 have been rewritten. -------------------Line 116-122

Page 5: The authors use “analyst” throughout the text. For example, page 5 line 205 “analyst SERS signal”. I am guessing they mean “analyte“ as in “SERS signal of an analyte”.

Response:

Thank the reviewer for the correction. We have changed all “analyst” to “analyte” in the manuscript when it means “a substance being identified and measured.”

Line 303 “in the insert of Figure” – what is figure #.

Response:

Thank the reviewer for the correction. it is Figure 6.------------line 312

The authors discuss the permissible levels of paraquat in mg/kg but report LOD in μM, how do these compare?

Response:

Thank the reviewer for the comment. We have added mg/Kg units (after conversion with M units) in parentheses of all the LOD in this manuscript.  For example, carbaryl (mw=201.22), 5uM=5x10-6 M=5x10-6 X 201.22 g/L = 1.006 mg/Kg (propose in aqueous solution)-------line 308, 321, 333, 344, 362

Lines 353-355: “based on that these pesticides can fluorescence imaging or not with respect to the nonfluorescent control contents in the solid-state”. I don’t understand the meaning of this statement.

Response:

Thank the reviewer for the comment. We defined the LOD of pesticide is the lowest concentration (the smallest amount) of an analyte that can be detected by using fluoresce microscopy. And this concentration of analyst is not fluorescent in control chip (without nanowire).thus this sentence became “based on the lowest concentration (the smallest amount) of pesticide, which is non-fluorescent in control chip (without nanowire), that can be detected by using fluoresce microscopy.”-----line 363-364.

Submission Date           10 December 2020

Date of this review         20 Dec 2020 00:48:37

Reviewer 3 Report

Authors reported 3D silver RCW nanostructures for Surface-enhanced Raman scattering (SERS) effects and plasmon enhanced fluorescence (PEF) sensing of different pesticides with micromolar limit of detection. The concept is fine, the performance is fine, probably the sensor was not fully optimized and some critical aspects for sensor have not been addressed. I hope that authors could perform additional experiments and improve their work

  1. Silver is prone to the oxidation, could you demonstrate how long the sensor is stable during storage?
  2. There are some typos: line 140 “(concentration ~ 2.6 OD, as shown in Figure 2a)”, what is OD?; Line 152 “of no specialinstructions.”
  3. The main question for me is the reproducibility of sensing, as you deposited silver RCW to substrate by drop casting, I suppose that deviation of SERS signal intensity will really vary, what is RSD on one sample and between sample for some pesticide?
  4. 2e,f : improve quality of text on figures
  5. 3c, please, mark concentrations with “M” (if it is molar concentrations)
  6. Please, mention the solvent for the preparation of analyte solutions
  7. Please, make a Table with peaks assignation from SERS spectra, it will help tracking SERS spectra
  8. 6d – spectra should be given until 2500 cm-1
  9. There are a number of other sensing system with lower LODs, where your sensor overperforms them?

(for example, 10.3390/nano7060142; 10.1016/j.aca.2019.03.058; 10.1039/C4AY03067B)

Author Response

Response to Reviewer 3 Comments

Comments and Suggestions for Authors

Authors reported 3D silver RCW nanostructures for Surface-enhanced Raman scattering (SERS) effects and plasmon enhanced fluorescence (PEF) sensing of different pesticides with micromolar limit of detection. The concept is fine, the performance is fine, probably the sensor was not fully optimized and some critical aspects for sensor have not been addressed. I hope that authors could perform additional experiments and improve their work

  1. Silver is prone to the oxidation, could you demonstrate how long the sensor is stable during storage?

Response:

Thank the reviewer for the comment. Yes,  metals can have oxidation problems, especially thiolation. This will cause the effect of SERS to deteriorate. That is why many commercial SERS chips or wafers are oxygen-free (vacuum) packaged. In our system, there is a coating layer of PVP on the Ag NWs, hence there is less problems about oxidation and thiolation (Also refer to reference 27 in manuscript). The related description in line 291-- “The SERS spectra obtained for R6G on a freshly prepared 3D-RCW composite film and that stored for 80 days (Figure 3b) without vacuum sealer storage revealed that the SERS intensity for R6G from the stored substrate was reduced less than 10%, indicating that the 3D-RCW composite film was quite stable when stored over a period of time.”

  1. There are some typos: line 140 “(concentration ~ 2.6 OD, as shown in Figure 2a)”, what is OD?; Line 152 “of no specialinstructions.”

Response:

Thank the reviewer for the correction. (1) In general, measuring the optical density (OD) is a common method to quantify the concentration of substances (Beer-Lambert law). It is usually the Y ordinate of the absorption spectrum. We rewrote this sentence as “the concentration is about 2.6 optical density (OD) in Figure 2a”-------------line 147. (2) “no specialinstructions “ have changed to “no special instructions “------line 159

  1. The main question for me is the reproducibility of sensing, as you deposited silver RCW to substrate by drop casting, I suppose that deviation of SERS signal intensity will really vary, what is RSD on one sample and between sample for some pesticide?

Response:

Thank the reviewer for the comment. Experiment result about Figure 3b can answer this query. In line 216- 218.

  1. 2e,f : improve quality of text on figures

Response:

Thank the reviewer for the comment. We replotted Figure 2 e,f to promote the text quality.

  1. 3c, please, mark concentrations with “M” (if it is molar concentrations)

Response:

Thank the reviewer for the correction. We replotted Figure 3c with the concentration unit is “M”.

  1. Please, mention the solvent for the preparation of analyte solutions

Response:

Thank the reviewer for the comment. In this manuscript, all analyte solutions are aqueous solutions, as described in line 150, 160, 285.

  1. Please, make a Table with peaks assignation from SERS spectra, it will help tracking SERS spectra

Response:

Thank the reviewer for the comment. We sorted out the SERS data of pesticides and listed in Table 1.-------------line 324

Table 1: Assignments of Raman modes in pesticides, taken from references as indicated in manuscript. SERS unit (cm-1).

Thiram

SERS           Mode

Carbaryl

SERS                   Mode

Paraquat

SERS                 Mode

Fipronil

SERS         Mode

567

C–S stretching

837

C-N stretching

1150

N=C=S stretching

1191

C=C bending

1386

C–N stretching

1382

symmetric vibration (naphthalene ring)

1293

C-C structural distortion

1516

C–H wagging

1442

C-H wagging of naphthalene ring

1578

stretching of C=C double

1642

C=N stretching

2253 

nitrile (−C≣N)

  1. 6d – spectra should be given until 2500 cm-1

Response:

Thank the reviewer for the comment. We have replotted the Figure 6d to set the x-coordinate to the range 2500 cm-1.

  1. There are a number of other sensing system with lower LODs, where your sensor overperforms them? (for example, 10.3390/nano7060142; 10.1016/j.aca.2019.03.058; 10.1039/C4AY03067B)

Response:

Thank the reviewer for recommending these references. We figure out as follows

  1. 3390/nano7060142: Nanomaterials 2017, 7(6), 142
  • A review paper……
  • Figure 1 and 2, three dimensions of Au nanostructure is better than one dimension.
  • Figure 3, flow-focusing microfluidic device with PDMS substrate.
  • Figure 4, Slippery liquid-infused porous surface-enhanced Raman scattering (SLIPSERS) with a micro/nanoporous substrate, high sensitivity.
  • Figure 5, SERS samples were prepared by mixing 0.5 ml of the Au nanoparticle suspension with 250 µl of RDX standard solutions at different concentrations. 15 mg/L LOD.
  • Figure 6, Immobilization of the nanoparticles on a solid substrate provides a means by which to bring nanoparticles into close proximity to one another, 25ppm LOD.
  • Figure 7, in one method of preparing SERS substrates on paper filters, clusters of Au NPs on the cellulose fibers of the paper. LOD 0.94 ng.
  • Figure 8, nanosphere lithography process for fabricating metal film over nanosphere (FON) to get periodic nanoparticle arrays as substrate. LOD 1 ppb.
  • Figure 9, Another method to fabricate flexible SERS substrates uses the roll-to-roll ultraviolet nanoimprint lithography (R2R UV-NIL) technique (with respect to Figure 8). A simple ‘paste and peel off’ sampling approach.
  • Figure 10, Electron-beam lithography techniques. The primary advantage of electron-beam lithography over other methods used to create SERS substrates is that it can draw custom patterns with sub-10 nm resolution.
  • Figure 11, lithographically substrate is comprised of an array of pyramidal shaped pits etched into silicon. Detect melamine by SERS and achieved a limit of detection of 33 ppb.
  • Figure 12, inexpensive SERS substrates, called P-SERS, that are comprised of gold nanoclusters on the surface of cellulose paper. LOD 10 ng.
  • Figure 13, chip elevated Au ellipse dimer array. Complicated sample preparation with chemical method. SERS analysis gave concentrations of 0.343 ± 0.025 mg L−1 perchlorate for sample CPMW-5 and 2.47 ± 0.16 mg L−1 perchlorate for sample CPMW-2D.
  • Figure 14, carboxylic acid terminated alkanethiol derivatized Au nanostars. Chemical modification on nanostructure surface. A detection limit of ~8 × 10−7 M was achieved.
  • Figure 15, lab-on-a-bubble (LoB) assay for SERS-based detection of an analyte. a detection limit of 173 ppt cyanide was achieved.
  • Figure 16, a SERS-active capture matrice with a magnet has been used to concentrate capture matrices onto an optical surface prior to detection of the analyte by SERS (Au/MEPH+ substrates).

  1. 1016/j.aca.2019.03.058: Analytica Chimica Acta, 2019, 1068, 70-79
  • Construction of Metal-organic framework (MOF-5), this substrate needed several chemical preparation processes.
  • Test organophosphorus pesticides.
  • LOD to 10-12 M with 10~30 mL of analyte solution.

  1. 1039/C4AY03067B- Anal. Methods, 2015, 7, 2563-2567
  • Substrate/method: The test solutions were mixed with this Ag sol gel + analysis program (algorithms)
  • The SERS spectra were collected by 100 mg/L pesticide (high concentration), and then progressing algorithms to find the fast and accurate determination of organophosphate pesticides.
  • No LOD discussion.

In order to display the high detection limit, most of these literatures fabricate their chip with rare material, complicated process or more than one chemical reaction. This cause cost- rising of chip. Our chip present not so high sensitivity, while the high lights are:

  1. Most literatures use Au, we use Ag but very stable.
  2. Most literatures use nanoparticle, we use nanowire.
  3. 3D substrate was constructed in our chip.
  4. Simple preparation of chip, Simple preparation of analytes.
  5. Most literatures detect just one and two pesticides, we detect four pesticides which belong to three type of pesticides (carbamate, paraquat and fipronil). Especially, fipronil is hard to detect by SERS.
  6. we can detect fluorescence and Raman by using the same chip.

We never want to defeat or overcome any study result in literature. Even in Table 2 of first review paper, they just summarized and compared the SERS substrates and analytes discussion.

Round 2

Reviewer 2 Report

The authors have addressed the majority of my comments. However, there are still some issues remaining:

1) Figure 1 still has "analyst" rather than "analyte" although it has been changed throughout the text.

2) line 257: "bigger aspect" should be "bigger aspect ratio".

3) Consistency in units is recommended throughout the text. The authors use mg/Kg but sometimes also mg kg-1.

Author Response

Comments and Suggestions for Authors

The authors have addressed the majority of my comments. However, there are still some issues remaining:

1) Figure 1 still has "analyst" rather than "analyte" although it has been changed throughout the text.

Response:

Thank the reviewer for the correction. We replotted Figure 1.

2) line 257: "bigger aspect" should be "bigger aspect ratio".

Response:

Thank the reviewer for the correction. We have revised in line 261.

3) Consistency in units is recommended throughout the text. The authors use mg/Kg but sometimes also mg kg-1.

Response:

Thank the reviewer for the correction. Whatever mg/kg, mgKg-1, We have revised to mg/Kg. in line 319, 333, 344.

Reviewer 3 Report

I appreciate comments of Author, most of the issues were solved, however, I still have some recommendations:

  1. Question 3. Please, provide the calculation strategy for the error calculations and also add the related content to Experimental Part
  2. Question 9. I recommend to prepare the Table about the comparison of the developed sensor and previously reported or add the brief paragraph about this (most of conclusion in this answer is fine), just better to make it look more systematic

Author Response

Comments and Suggestions for Authors

I appreciate comments of Author, most of the issues were solved, however, I still have some recommendations:

  1. Question 3. Please, provide the calculation strategy for the error calculations and also add the related content to Experimental Part

Response:

Thank the reviewer for the comment. We add the description about the calculated methods of repeatability and stability of 3D RCW nanochip in 2.5. Raman measurement of 2. Experiment.----line 162-166.

Data reproducible repeatability of 3D RCW nanochip: SERS spectra was collected from seven different places of chip and then averaged, as the standard spectrum. The algorithm of data deviation comes from the intensity-subtraction between every spectrum and standard spectrum. The stability of 3D RCW nanochip: these SERS signals can still be detected and the intensity is less than 10 percentage, after a chip was placed in the atmosphere for several days.

  1. Question 9. I recommend to prepare the Table about the comparison of the developed sensor and previously reported or add the brief paragraph about this (most of conclusion in this answer is fine), just better to make it look more systematic

Thank the reviewer for the comment. We summarized the Table 2 according to comparisons between this method and literatures supported by Reviewer. We also add the answer and description and in the manuscript. We also fulfilled the related references in Table 2. ----line 378-393.

Table 2. Comparison of substrates and LODs of this method with literatures

Analytes

LODs

Substrates

Nobel metal nanostructure ( Exciting laser)

Refs

rhodamine 6G

(R6G)

-

SERS signal: nanostars > nanotriangles> nanospheres

gold nanostructures

(785 nm lasering)

43

cyclotrimethylenetrinitramine (RDX)

0.15

mg/L

glass slide

Au NP suspension

(785 nm lasering)

44

methamphetamine

-

flow-focusing microfluidic (PDMS)

controlled Ag-NP/salts aggregation

(633 nm lasering)

45

DNA bases

4-aminothiophenol (4-ATP)

1 fM-

10 aM

Slippery liquid-infused porous surface-enhanced Raman scattering (SLIPSERS)

Au NPs +analyte

(633 nm lasering)

46

polychlorinated

biphenyls (PCBs)

1PPb

Naphthalene

25 ppm

“chemical tether” to anchor NPs on a quartz substrate

Au NPs

(633 nm lasering)

47

2,4,6-trinitrotoluene (TNT), 2,4-dinitrotoluene (2,4-DNT), 1,3,5-trinitrobenzene (TNB)

0.89~

94 pg.

clusters of NPs on the cellulose fibers of the paper.

Au NPs

(785 nm lasering)

48

melamine

1 ppb

physical vapor deposition and then electrochemical deposition

(formation of Ag or Au metal film over nanosphere (FON) surface)

(785 nm lasering)

49

parathion-methyl thiram

chlorpyrifos

2.60

0.24

3.51 ng/cm2

Nanoparticle solution was

dropped uniformly on the sticky side of  adhesive tape

Au NPs (paste and peel off’ sampling approach.)

(633 nm lasering)

50

crystal violet

Mitoxantrone

10 nM

1 nM

Electron Beam Lithography and Nanotransfer Printing

Au nanopattern

(633 nm lasering)

51

melamine

33 ppb

lithographically substrate is comprised of an array of pyramidal shaped pits etched into silicon.

Ag/Au NP spyramidal Klarite@ substrates

52

organophosphate malathion, heroin,

cocaine.

413 pg,

9 ng,

15 ng

Inkjet-printed paper-based dipsticks and swabs

Au nanoclusters

(785 nm lasering)

53

Perchlorate.

0.343 ± 0.025 mg L−1

Electron beam lithography

Au ellipse dimer array

(785 nm lasering)

54

Uranyl solution

120 nM.

Chemical modification on nanostructure surface.

Au nanostars.

(785 nm lasering)

55

cyanide

173 ppt

lab-on-a-bubble (LoB) assay

Au NP-coated LoBs

56

hexavalent chromium

59 ppb

a capture matrice with self-assembled monolayer on

Immobilizing nanocluster

Au/MEPH+ substrates

(785 nm lasering)

57

Paraoxon

Fenitrothion

10-12 M

Metal-organic framework (MOF-5)

Au-grating

(785 nm lasering)

58

methyl parathion,

edifenphos,

ethyl paraxon

*

Spectra collection , then chemometric methods (standard normal variate variance (SNV)method)

*improve the discriminant accuracy

Ag sol gel

(532 nm lasering)

59

  1. Tian, F.; Bonnier, F.; Casey, A.; Shanahan, A.E.; Byrne, H.J. Surface Enhanced Raman Scattering with Gold Nanoparticles: Effect of Particle Shape. Anal. Methods 2014, 6, 9116–9123.
  2. Hatab, N.A.; Eres, G.; Hatzinger, P.B.; Gu, B. Detection and Analysis of Cyclotrimethylene-trinitramine (RDX) in Environmental Samples by Surface Enhanced Raman Spectroscopy. J. Raman Spectrosc. 2010, 41, 1131–1136.
  3. Andreou, C.; Hoonejani, M.R.; Barmi, M.R.; Moskovits, M.; Meinhart, C.D. Rapid Detection of Drugs of Abuse in Saliva using Surface Enhanced Raman Spectroscopy and Microfluidics. ACS Nano 2013, 7, 7157–7164.
  4. Yang, S.; Dai, X.; Stogin, B.B.; Wong, T.-S. Ultrasensitive Surface-Enhanced Raman Scattering Detection in Common Fluids. Proc. Natl. Acad. Sci. USA 2016, 113, 268–273.
  5. Péron, O.; Rinnert, E.; Lehaitre, M.; Crassous, P.; Compère, C. Detection of Polycyclic Aromatic Hydrocarbon (PAH) Compounds in Artificial Sea-Water using Surface-Enhanced Raman Scattering (SERS). Talanta 2009, 79, 199–204.
  6. Fierro-Mercado, P.M.; Hernández-Rivera, S.P. Highly Sensitive Filter Paper Substrate for SERS Trace Explosives Detection. Int. J. Spectrosc. 2012, 2012, 716527.
  7. Wang, J.F.; Wu, X.Z.; Xiao, R.; Dong, P.T.; Wang, C.G. Performance-Enhancing Methods for Au Film over Nanosphere Surface-Enhanced Raman Scattering Substrate and Melamine Detection Application. PLoS ONE 2014, 9, e97967.
  8. Chen, J.; Huang, Y.; Kannan, P.; Zhang, L.; Lin, Z.; Zhang, J.; Chen, T.; Guo, L. Flexible and Adhesive Surface Enhance Raman Scattering Active Tape for Rapid Detection of Pesticide Residues in Fruits and Vegetables. Anal. Chem. 2016, 88, 2149–2155.
  9. Abu Hatab, N.A.; Oran, J.M.; Sepaniak, M.J. Surface-Enhanced Raman Spectroscopy Substrates Created via Electron Beam Lithography and Nanotransfer Printing. ACS Nano 2008, 2, 377–385.
  10. Smith, E.; McNay, G.; McInroy, A.; Fitchett, K. New Developments in SERS for the Pharmaceutical Industry. 2017,2.
  11. Yu, W.W.; White, I.M. Inkjet-Printed Paper-Based SERS Dipsticks and Swabs for Trace Chemical Detection. Analyst 2013, 138, 1020–1025.
  12. Jubb, A.M.; Hatzinger, P.B.; Gu, B. Trace-Level Perchlorate Analysis of Impacted Groundwater by Elevated Gold Ellipse Dimer Nanoantenna Surface-Enhanced Raman Scattering. J. Raman Spectrosc. 2017, 48, 518–524.
  13. Lu, G.; Forbes, T.Z.; Haes, A.J. SERS Detection of Uranyl using Functionalized Gold Nanostars Promoted by Nanoparticle Shape and Size. Analyst 2016, 141, 5137–5143.
  14. Schmit, V.L.; Martoglio, R.; Scott, B.; Strickland, A.D.; Carron, K.T. Lab-on-a-Bubble: Synthesis, Characterization, and Evaluation of Buoyant Gold Nanoparticle-Coated Silica Spheres. J. Am. Chem. Soc. 2012, 134, 59–62.
  15. Mosier-Boss, P.A.; Putnam, M.D. Detection of Hexavalent Chromium using Gold/4-(2-Mercaptoethyl)pyridinium Surface Enhanced Raman Scattering-Active Capture Matrices. Anal. Chim. Acta 2013, 801, 70–77.
  16. Guselnikova O, Postnikov P, Elashnikov R, Miliutina E, Svorcik V, Lyutakov O. Metal-organic framework (MOF-5) coated SERS active gold gratings: A platform for the selective detection of organic contaminants in soil. Anal Chim Acta. 2019, 1068, 70-79.
  17. Weng, S. Li, M. Chen, C. Gao, X. Zheng, S. Zeng, X. Fast and accurate determination of organophosphate pesticides using surface-enhanced Raman scattering and chemometrics. Anal. Methods 2015, 7 (6) , 2563-2567.

This manuscript is a resubmission of an earlier submission. The following is a list of the peer review reports and author responses from that submission.

Round 1

Reviewer 1 Report

In the present manuscript, results related to the design, fabrication and characterization of silver nanowire and 3D random crossed‐wire woodpile (RCW) nanostructures are reported. The authors propose to use these nanostructures for Plasmon-Enhanced Fluorescence (PEF) and Surface-Enhanced Raman Scattering (SERS) measurements.  PEF and SERS tests were performed on Rhodamine 6G. Further tests were reported for four different kinds of pesticides.

The topic addressed by the authors is interesting due to the growing relevance of  PEF and SERS in analyte detection, even though the proposed method is not particularly innovative. In addition, the manuscript presents many points that need to be carefully addressed by the authors starting from the title. What synergistic effect do the authors intend to emphasize? Usually, when good quality Raman and SERS measurements are to be obtained, the fluorescence must be kept under control. The authors try to explain the relationship between PEF and SERS in the Introduction, but this section results extremely confused. Some statements are really difficult to understand (e.g. see lines 33-34, 81-83 and others).

In the Experiment Section (better to name it Materials & Methods), many details are missed. In particular, no information about the Raman set-up is given. In particular, the laser wavelength is not reported and, as a consequence, the reader can not have any idea about the occurrence of fluorescence and/or Raman scattering that is one of the points that the manuscript should discuss.

Given the lack of details in the Experiments Section, many aspects of the reported results are also unclear. Some Figures need a more precise discussion. For all these reasons, the paper cannot be accepted for publication on Molecules in the present form. A very deep revision is strictly required.

A not exhaustive list of further points that need to be revised is reported below:

  • a.Please check lines 16-17, 185, 267, 276, 283, 291-292, Fig. 3, Fig. 6 caption, line 312. Concentrations cannot be given in mole units, molars have to be used, instead.
  • b. Line 86: the sentence “producing intensities that are orders of magnitude greater than the incident light intensity” is a misunderstanding. Please notice that the electrical field is concentrated near the metal surface, not the light intensity.
  • In the Materials section, the authors should report the analytical grade of the employed chemicals.
  • Please check the details given for the spectrofluorometer and cuvette al lines 100-102.
  • As far as concerns AFM measurements, please check the sentence reported al line 103.
  • As said before, no detail of the Raman spectroscopy apparatus is given. The excitation light wavelength is essential information for understanding the interaction with nanostructures. Usually, nanoparticle dimensions are suitably tuned to a specific wavelength to maximize the SERS performance. Please, discuss this important aspect. Please, report the spectral resolution of the Raman apparatus too.
  • Please note that ref. 11 is not pertinent (see line 108).
  • What water was used for removing excess EG and PVP? (See line 123).
  • At line 138 the authors affirm “The silver nanoparticles and nanowires were synthesized “. Are silver nanoparticles also produced? Please clarify this point.
  • What does dd water mean at line 139?
  • Please check the sentence at lines 176-177.
  • Please check line 189.
  • What excitation and detection wavelengths were used for Figure 5?
  • Please carefully revise the discussion of Figure 6. The reported insets have to be discussed in terms of concentration linear range and line fitting, limit-of-detection and sensitivity.
  • If also PEF is proposed for sensing pesticides, quantitative data should be given (concentration linear ranges and line fitting, limit-of-detection and sensitivity).
  • The authors should add their previous work (Faraday Discuss., 2017, 196, 55) in the References list in order to give additional information on the proposed method for nanostructure fabrication.
  • Please check the manuscript for misprints and English language misuse (e.g. line 320 and others).

Author Response

Dear Assistant Editor:

Dr, Aaron Yan

Enclosed please find a copy of the revised manuscript entitled “Synergistic effect based on a 3D-nanostructure plasmon-enhanced fluorescence and surface-enhanced Raman scattering detection” for publication in Molecules.

We deeply appreciate you and referee’s kindness and patiently recommend us about the manuscript. The editor and reviewers’ comments are addressed in full as possibly as we can. First, we added several experimental results (Figure 2e, 2f and Figure 7) and redrawn Figure 6 and discussed some issues follow the suggestions from reviewers. Second, we corrected the mistakes, fulfilled some references for this time revision. All defenses and revisions are marked with blue color in manuscript and illustrated as follows.

I hope that you find this manuscript acceptable for publication in the article of Molecules. Please inform me if you have any other query and request.

Best regards,

Cheng-Chung Chang 2020 11 30

8864-2284-0734#24,

ccchang555@dragon.nchu.edu.tw

Reviewer 2 Report

Response:

In this article the authors claim in a work on the synthesis of silver nanowires with applications in Surface-Enhanced Raman Scattering, SERS, and Plasmon-Enhanced Fluorescence, PEF. It is therefore worthy of publication; however, there are some indications that must be addressed to the manuscript.

  1. In line 28, the authors say: The SPR character strongly depends on the noble metal species, size and shape of the nanostructures. The size and shape of the nanostructures are characteristic mainly of the Localized Surface Plasmon Resonance, otherwise they are spr not localized and depends mainly on the thickness of the metal layer. Many nano structures have been used to enhancement the signal phenomena such as SERS or PEF, such as nano-stars, nano-rods, nano-shells, nano-cubes, etc. Are there advantages of the authors' proposal in this work with respect to the other nanostructures?
  2. In line 150, the authors say: “Furthermore, Figure 2b and c present the microscopy, AFM and SEM images showing that the average values for the diameter and length of the Ag NWs for all surfaces was 80-120 nm and >10 μm, respectively. The average aspect ratio (length/diameter) of the nanowires is more than 100. The thickness of the PVP coating on the surfaces of the nanowires was measured to be 20–25 nm.” How do the authors estimate the dimensions of the nano wires? with the scale bars?
  3. In line 154, The authors say: “Figure 2. (a) Absorption spectra for Ag NWs.” The measurement is the absorption or absorbance? in what units?
  4. In line 225, the authors say: “Figure 5. Hot spot distribution for 3D-RCW at the intersection of nanowires (a) and both ends of the nanorods (scale bar: 10 μm)”. it remains to describe the b)
  5. In line 236, The authors say: Thus c.a. 200 nm thick 3D hotspot network is optimal. In our case, it is reasonable that 3 cycles of Ag NW spreading is thick enough to permit measurement of the optical properties and SERS performance as well as the penetration depth limitation for visible light. How do the authors estimate the dimensions of the nano wires? with the scale bars? what does c.a. mean?
  6. In line 250, the authors say: The SERS spectra obtained for R6G on a freshly prepared 3D-RCW composite film and that stored for three months (Figure 3b) without vacuum sealer storage revealed that the SERS intensity for R6G from the stored substrate was reduced less than 10%, Silver nano wires do not show a decrease in plasmonic response due to oxidation of silver, something very usual for this material?

Author Response

(The authors gave the same response as above.)

Round 2

Reviewer 1 Report

In my opinion, the present version of the manuscript presents some improvements, but some points have not been correctly addressed by the authors and the paper remain unclear in many aspects. They do not clarify what synergistic effect they want to emphasize. In the Introduction and the new 3.4 paragraph, they modified previous statements but some sentences are still unclear. The reasons given for justifying the use of the proposed substrates for PEF and SERS are not convincing. They still want to use mole for reporting their results, but their choice avoids comparison with other results available in the literature.

For all these reasons the manuscript cannot be accepted for publication and should be rejected.